# Student Perspectives on the Pharmacist’s Role in Deprescribing Opioids: A Qualitative Study

**DOI:** 10.3390/pharmacy11040116

**Published:** 2023-07-13

**Authors:** Alina Cernasev, Rachel E. Barenie, Sydni Metzmeier, David R. Axon, Sydney P. Springer, Devin Scott

**Affiliations:** 1Department of Clinical Pharmacy and Translational Science, College of Pharmacy, University of Tennessee, Health Science Center, Nashville, TN 37211, USA; acernase@uthsc.edu (A.C.); smetzmei@uthsc.edu (S.M.); 2Department of Pharmacy Practice & Science, College of Pharmacy, University of Arizona, Tucson, AZ 85721, USA; draxon@arizona.edu; 3Department of Pharmacy Practice, University of New England School of Pharmacy, Westbrook College of Health Profession, Portland, ME 04013, USA; sspringer1@une.edu; 4Teaching and Learning Center, Department of Academic, Faculty and Student Affairs, University of Tennessee Health Science Center, 920 Madison, Suite 424, Memphis, TN 38163, USA; dscott50@uthsc.edu

**Keywords:** deprescribing, opioids, USA

## Abstract

Introduction: Opioid over-prescribing has led to changes in prescribing habits and a reduction in the amount of opioid prescriptions per patient. Deprescribing has proved to be an effective way of decreasing the number of opioids patients are receiving, and pharmacists are in the optimal position to provide these services for their patients. However, student pharmacists require additional education and training to be able to understand their role in deprescribing opioids upon entering the profession. Methods: Student pharmacists at three United States of America schools of pharmacy were invited to participate in virtual focus groups about deprescribing opioids in Fall 2021. A trained qualitative researcher conducted the focus groups, which were audio-recorded and later transcribed verbatim for thematic analysis. Two independent qualitative researchers coded the transcripts using both inductive and deductive approaches. The researchers then met to identify, discuss, and describe themes from the data. Results: Thematic analysis revealed two themes: (1) perceived obstacles and enablers to initiate deprescribing for opioid medications and (2) additional pharmacy curricula experiences are necessary to better equip student pharmacists to address deprescribing. These themes emphasize the challenges student pharmacists face as well as opportunities to enhance their knowledge to be practice-ready. Conclusion: Varying educational approaches to teaching deprescribing in the pharmacy curriculum, including objective structured clinical exams, interprofessional education, and motivational interviewing, should be further assessed.

## 1. Introduction

Three decades ago, healthcare prescribers in the United States of America (USA) began over-prescribing opioids for non-cancer-related chronic and severe pain, which resulted in opioid overuse and misuse [1]. Opioid misuse and overuse continues to be a problem. For example, in 2016, Tennessee was ranked as one of the states with the highest opioid prescription rate per capita, accounting for 107.5 opioid prescriptions per 100 residents [2]. While prescribers have changed their opioid prescribing habits in recent years, in conjunction with other efforts to alleviate the problem of opioid over-prescribing in the USA, more progress is needed to combat the opioid epidemic [3].

Many of the efforts to combat opioid overuse and misuse are aimed at avoiding the inappropriate initial prescription of opioids. For example, the Centers for Disease Control and Prevention (CDC) has published and continues to update their national guidelines for opioid prescribing to advance the use of non-opioid and non-pharmacological treatment options for managing chronic non-cancer-related pain first before starting an opioid medication [4]. Most states have implemented a Prescription Drug Monitoring Program which allows for real-time monitoring of opioid prescriptions and other controlled substances dispensed to patients [5]. Some states, such as Tennessee, even have legislated limits on the number of days’ supply of opioids or milligram morphine equivalents that may be prescribed to patients [6]. While these measures are important, promoting “deprescribing” is also necessary. This concept of “deprescribing” is a patient-centered approach to slowly decreasing and/or stopping an opioid medication to improve health outcomes for the patient by reducing the long-term exposure to harmful effects of opioids such as fractures, poor physical functioning, fatigue, and sleep disturbances [7,8]. The CDC guidelines on opioid prescribing recommend developing a plan for tapering-off and discontinuing opioids when they are no longer necessary or the risks outweigh the benefits of continued use, which provides support for deprescribing efforts [4].

Prior research has shown that deprescribing is an effective approach to improve patient outcomes that involves all members of the healthcare team. For example, a systematic review of deprescribing dual-purpose medications in older adults concluded that deprescribing can lead to improved mortality and decreased acute care visits [9]. Further studies have reported similar outcomes regarding the safety, ease, and value of deprescribing [10]. More specifically to opioids, deprescribing can decrease the adverse effects, risk of misuse, and physical dependence related to opioid use. In recent years, research has also found that patients are more accepting of deprescribing efforts. Additionally, patients are more agreeable to deprescribing when they have a trusting relationship with a provider [11,12]. These efforts are not without their limitations, such as concerns about patients seeking other alternatives to self-manage their pain if not appropriately prescribed [13,14]. Deprescribing can play an important role in improving the health outcomes of patients on opioid therapy.

Pharmacists, specifically, can play an integral role in deprescribing efforts due to their extensive drug knowledge regarding therapeutic uses of medications like opioids and their unique position as the most accessible healthcare providers. Prior research has shown that pharmacist-initiated deprescribing efforts are feasible and may positively impact patient outcomes [15]. Additionally, through the use of Collaborative Pharmacy Practice Agreements (CPPAs), pharmacists may be provided the prescriptive authority to start and/or stop medications so long as it is within the scope of their agreement and in compliance with federal and state law [16,17]. In practice, this means that pharmacists could manage medications for patients with chronic pain by stopping inappropriately prescribed opioids (or tapering patients off the opioid—deprescribing) and starting alternative therapies when appropriate or necessary [18]. However, researchers have found that student pharmacists lack confidence or are unprepared to engage in deprescribing efforts in practice [19]. While pharmacists are well positioned to approach opioid deprescribing with their patients, further research is needed to identify and operationalize opportunities to increase student confidence in deprescribing.

In sum, existing data suggest that deprescribing is evidence-based, pharmacists have a key role to play in deprescribing, and additional education and training on the topic may be necessary. Little is known about student pharmacists’ perspectives on deprescribing opioid medications specifically and their readiness to do so. Yet, pharmacy students must be prepared to collaborate with other healthcare professionals to recommend evidence-based modifications to medication therapy and possibly even start or stop treatment under the authority of a CPPA upon entering the profession. The goal of this study was to further examine how pharmacy students perceive their role in deprescribing opioids for their patients.

## 2. Materials and Methods

### Study Design and Data Collection

A qualitative focus group (FG) approach was deemed appropriate as it is well suited for capturing participants’ perspectives who were geographically located in different parts of the USA (University of Arizona, University of New England, University of Tennessee Health Science Center) and provides detailed descriptions of a complex phenomenon, deprescribing of opioids [20,21].

The Doctor of Pharmacy (Pharm.D.) curriculum at all three colleges of pharmacy (UTHSC, UNE, and the University of Arizona) is a four-year program composed of didactic classes and experiential education. Student pharmacists study pharmacotherapy, medicinal chemistry, and pharmacology of opioids throughout the Pharm.D. curriculum.

The proposal for this research study was approved by the University’s Institutional Review Board (IRB) for each researcher involved in this study, including the University of Tennessee Health Science Center (IRB # 21-08234-XM, 2 July 2021), University of New England (IRB # 0821, 2 August 2021), and University of Arizona (IRB# 2021-015-PHPR, 9 August 2021).

All pharmacy students currently enrolled in their programs at three colleges of pharmacy at the time of the study were eligible to participate in this study. The participants in this qualitative study were identified via an email sent to the respective colleges of pharmacy where they were enrolled in Fall 2021. Participants had the opportunity to participate voluntarily in this study. The FG instrument was developed by the researchers based on an extensive review of the literature and the Theory of Planned Behavior (TPB) [22]. Demographic data were collected at the start of each FG. The instrument contained questions asking about general aspects of deprescribing, such as how much training on deprescribing the students had received in their pharmacy curriculum, how detailed the training was, which classes or years this was provided in, what experiences the students had encountered with deprescribing, and how deprescribing was assessed in their curriculum. These are described in more detail in previous publications [23,24]. Some of the questions asked included the following:What experiences have you encountered with opioid deprescribing?What training have you received on deprescribing in your curriculum?How could training on deprescribing be improved in the curriculum?How comfortable are you initiating deprescribing or making a recommendation to initiate deprescribing to another member of the healthcare team?

The instrument also contained more specific questions around certain aspects of deprescribing, in particular, the deprescribing of opioids. Opioids were of interest to the research team given the opioid epidemic in the US and the potential role of pharmacists in helping to address inappropriate opioid use. This manuscript focuses on questions and responses about deprescribing opioids. Figure 1 presents an overview of the study design, data analysis, and the results.

All FGs were audio recorded and transcribed verbatim by a third party to avoid bias. Identifying information was removed from the transcribed FG to protect participants’ confidentiality. Using thematic analysis, the foundational method for qualitative research, the FGs were coded accordingly and focused on the critical aspects of the FG instrument [25,26]. Consequently, this study used a symbiosis of deductive and inductive coding. Thus, the TPB framework facilitated the deductive codes, while the thematic analysis approach enabled the inductive codes to be indexed and the relationships between categories were examined systematically [22,25].

Data (FG transcripts) were analyzed systematically using both inductive and deductive approaches by two researchers [22,25]. Dedoose^®^ (Manhattan Beach, CA, USA), a qualitative analysis software, facilitated hierarchical or tree-like coding [27,28]. After researchers independently analyzed the data, they then convened to discuss, critically describe, analyze, and justify identified themes. A subsequent consensus meeting was held to resolve coding discrepancies. Data were collected and analyzed until saturation was reached [29,30].

## 3. Results

A total of 1366 student pharmacists from three colleges of pharmacy were invited to participate voluntarily in the virtual FG. Twenty-six student pharmacists participated in four FGDs. Of 26 participants, 14 self-identified as male, 2 declined to state their gender identity, 16 self-identified as White, 4 as Asian, 2 as Latino, 1 as Middle Eastern, 1 as Pacific Islander, and 1 declined. Out of the 26 participants, 16 were enrolled in the fourth year, 7 were enrolled in the third year, and 3 were enrolled in the second year.

Two major themes emerged from the data analysis. The first theme presented the students’ perceived obstacles and enablers to initiating opioid deprescribing. The second theme explored recommendations to empower future generations of student pharmacists by implementing changes to the pharmacy curriculum that equip them with information necessary to support deprescribing efforts on the healthcare team.

### 3.1. Theme 1: Perceived Obstacles and Enablers to Initiating Opioid Deprescribing

The first emergent theme described various obstacles and potential enablers faced by practicing pharmacists to promote opioid deprescribing efforts, including relationships with patients, a synergistic collaboration with physicians, and the necessity for collaborative pharmacy practice agreements. Student pharmacists viewed relationships with patients as both a potential obstacle to and enabler for deprescribing opioids, as they identified successfully initiating deprescribing as being contingent upon the interpersonal relationship between an individual pharmacist and their patient. Likewise, student pharmacists asserted that relationships between pharmacists and physicians could serve as an enabler for or obstacle to deprescribing. Finally, collaborative pharmacy practice agreements were seen as a necessity for sustainable efforts to deprescribe opioids.

Several participants shared that situations exist when the pharmacist–patient relationship is essential to discontinuing an opioid medication. Additionally, the following excerpt highlights the patient concerns that students must overcome when having the discussion about discontinuing their opioid medication: 

“…I think that, when it comes to being a student…I’m trying to get people off of it [opioids], you know, but I don’t think that really goes over well because the patients a lot of times are really antsy to get the medication. So, I don’t feel like they’re always very receptive to deprescribing recommendations or alternative therapy recommendations.”(FG4, S1)

The following statement captures a growing sense of the importance of a relationship between patient and pharmacist and the fear of dismantling that well-developed relationship by introducing deprescribing. The student states:

“…But another thing is, like jumping back to what [FG3, S4] said is that you have a relationship with that patient, and you don’t want to ruin that relationship just for even thinking that they might be using their pain meds the wrong way. So just that not wanting to ruin that relationship and still being professional at the same time can be kind of scary.”(FG3, S3)

A shared collective view was that in order for a pharmacist to overcome the barrier of deprescribing, a synergistic relationship between the patient, physician, and pharmacist must be cultivated. Both the patient and physician must trust their pharmacist. This participant highlighted the important concern that patients may be scared to make this change because of the pain or the fear of withdrawal symptoms they may experience.

“… I think sometimes it’s hard to deprescribe in general that you get sent over from prescribers because more often than not, patients have a better relationship with their doctor than they do with their pharmacist. Not all the time, but sometimes. But I think, for opioids, it’s also difficult because it is a pain medication. Most likely your patient is in pain, or they may even be addicted to the medication, so if you try to tell them that you’re going to take them off, it might scare them.”(FG1, S3)

One of the FG discussions emphasized that there should be a strong partnership between pharmacists and physicians in order to maximize the likelihood of success when deprescribing, so that the physician and pharmacist can coordinate a slow decrease in the opioid medication before ultimately stopping the opioid treatment completely.

“…Deprescribing opioids…I don’t think that should fully rely on the pharmacist. I think that should be a shared responsibility between the pharmacist and the physician and an interprofessional conversation that needs to occur in order to do like a stepdown therapy.”(FG3, S4)

Several participants asserted the value of CPPAs to initiate the discontinuation of a medication, in this case an opioid. One participant summarized the importance of CPPA for a practicing pharmacist:

“…I think collaborative practice agreements gives pharmacists like the trust that doctors—like it empowers pharmacists because doctors trust them with their knowledge and with their practice. And I think another thing that could possibly empower pharmacists is just, you know, having the confidence that they know what they know and that they have the knowledge to do so kind of as a pharmacist, as well.”(FG3, S3)

Another FG discussion emphasized the importance of collaborating with the healthcare team, especially physicians. The following quote provides a student pharmacist’s opinion on how physicians who are recent graduates may be more willing to engage in a CPPA with a pharmacist. 

“Some states are making a collaborative practice legal… and I feel like maybe medical providers don’t want to see them sharing that power sometimes… I feel like a lot of the younger doctors and residents, they are really up for that because that’s how the way of teaching is going now, it’s really interprofessional.”(FG1, S1)

Participants also highlighted that different healthcare models could be used to develop the CPPA for opioid deprescribing, which was extracted from the FG discussions provided below.

“I did a previous rotation at the VA, deprescribing was very common there, actually, based on a bunch of different scenarios. One more recent, though, I was at a local clinic here in Tucson, Arizona, where the pharmacists actually operate under a collaborative practice agreement, and my preceptor was a pain management specialist, so we definitely did a lot of deprescribing, changing meds and all of that sort.”(FG1, S4)

“…Or maybe someone’s dose is too high on their blood pressure medication, and… they go to the pharmacist to discuss that, a pharmacist may want to go to the physician to change it unless you have a collaborative agreement in place.”(FG3, S5)

Student pharmacists identified several interrelated barriers to and enablers of opioid deprescribing. Participants saw weak or nonexistent relationships between patients and pharmacists or pharmacists and physicians as barriers to successful deprescribing. Conversely, participants viewed strong relationships between patients and pharmacists and pharmacists and physicians as potential aids to combatting the opioid epidemic through deprescribing. Student pharmacists identified the value in formalizing and improving the relationship between pharmacists and physicians through collaborative practice agreements, which they repeatedly pointed to as an integral part to successful opioid deprescribing. 

### 3.2. Theme 2: Additional Pharmacy Curricula Experiences Are Necessary to Better Equip Student Pharmacists to Address Deprescribing

This theme emphasizes how students may feel uncomfortable initiating a discussion with a patient about opioid medications and possible discontinuation of that medication, due in large part to a lack of formal and informal educational experiences. The theme presents various recommendations such as an objective structured clinical exam (OSCE), interprofessional education, and/or motivational interviewing to train students more effectively on deprescribing. 

Several participants expressed their hesitancy in communicating with a patient on how to decrease the dose of an opioid medication or even to begin discussions surrounding deprescribing. While a conversation about opioid deprescribing is most often centered on evidence-based guidelines, the anxiety surrounding it may also point to a fear of judgment. The following quotation calls attention to the discomfort a student pharmacist may experience when starting a conversation with a patient about opioid treatment and eventual discontinuation or change in the medication regimen. The participant uses the word “taboo” which emphasizes a sense of a forbidden discussion about the medication usage. 

“…I think everything what we just talked about before with opioids is a big gap because I feel like it’s almost taboo to talk about getting people off of their opioids because you are essentially telling someone, I am going to put you in pain, and I don’t think we were versed very well in having those conversations with patients.”(FG4, S4)

To become more comfortable initiating a discussion with a patient who is taking any opioid medication, several participants recommended more exposure through OSCE or other simulations in the pharmacy curriculum. During the FG discussions, there was a consensus regarding the need to incorporate more simulations that explore communication approaches for deprescribing opioid medications in the pharmacy curriculum.

“…I think something that we do at the school is we have like a simulation where you can call and leave a voicemail for the prescriber…like making an intervention in terms of opioid therapy… OSCE-style simulation. I think we don’t do enough of that. Because we need that type of experience…we need more opportunities, where like, hey, you have to pick up on it, but you also need to figure out, well, how do I word this, this would be really helpful.”(FG4, S10)

The following excerpt emphasizes the need for incorporating additional training in the curriculum on how to initiate the discussion with the patient to build their confidence. While the student felt that the importance of deprescribing was adequately covered in the curriculum, they recommended more emphasis on conversation skills surrounding deprescribing.

“…I personally feel like we have the knowledge on what opioid medications should be deprescribed. I think where we need more of the training is how to have those narratives. And I think that kind of comes from maybe a barrier of trust that they have with us, so maybe building that relationship more with the patient. And maybe that kind of falls into the motivational interviewing training that we are getting and stuff like that, but also, I just feel like we need better education and maybe more practice on having those conversations with patients and providers, who do prescribe these medications.”(FG4, S5)

One student points out how previous exposure to a particular population might play a role in interacting with patients. The same student also suggested that an interprofessional activity focusing on how to approach opioid deprescribing with other healthcare students’ teams may be beneficial.

“…As far as the opioids, that is a harder conversation to actually have with the patients, but it comes down to experience…also an interprofessional activity…with med students, nursing students…that we can call them and maybe let them know, you know, hey, this person, we can eliminate a medication for them that’s not helping them, it’s not been shown in literature to help, and it may even cause some harm.”(FG1, S4)

These recommendations are directly tied to theme 1: perceived obstacles and enablers to initiating deprescribing for opioid medications. Student pharmacists viewed enhanced education surrounding motivational interviewing and opioid deprescribing as beneficial to the relationship between patients and pharmacists. Furthermore, interprofessional education relating to opioid deprescribing was put forward as an educational activity that could foster positive partnerships between pharmacists and physicians. Lastly, OSCEs focused on opioid deprescribing were proposed to improve the relationships between patients and pharmacists and pharmacists and physicians. Participants offered recommendations for improving the pharmacy curriculum surrounding opioid deprescribing that directly addressed the perceived obstacles and enablers to initiating deprescribing highlighted in theme 1.

## 4. Limitations and Future Research

Some caveats must be noted when interpreting the present findings. Notably, the qualitative nature of this study may limit the generalization of the results to broader student pharmacist opinions.

Subsequently, these findings may benefit from future research that accounts for a broader interprofessional healthcare student narrative of developing and implementing collaborations to initiate deprescribing of opioids to improve patient outcomes. Complementary future studies on interprofessional partnerships in clinical environments would provide valuable insights into designing OSCEs and other pedagogical activities that resemble clinical settings. Nevertheless, the student pharmacist experiences representing the three different colleges of pharmacy featured in this study provide valuable feedback regarding the need to consider how educational activities focused on deprescribing opioids could be designed to mimic the clinical practice.

## 5. Conclusions

This study contributes to the literature by revealing two main themes: (1) perceived obstacles and enablers to initiate deprescribing for opioid medications and (2) additional pharmacy curricula experiences are necessary to better equip student pharmacists to address deprescribing. Student pharmacists asserted that strong relationships between pharmacists and patients, partnerships between pharmacists and physicians, and collaborative pharmacy practice agreements are key to successful opioid deprescribing. Furthermore, participants recommended that schools of pharmacy integrate more opioid deprescribing-focused OSCEs, interprofessional education, and motivational interviewing into the curriculum to improve pharmacist comfort with and the ability to safely deprescribe opioids. These themes emphasize the challenges student pharmacists face as well as opportunities to enhance their knowledge to be practice-ready. Varying educational approaches to teaching deprescribing should be further assessed.

## Figures and Tables

**Figure 1 pharmacy-11-00116-f001:**
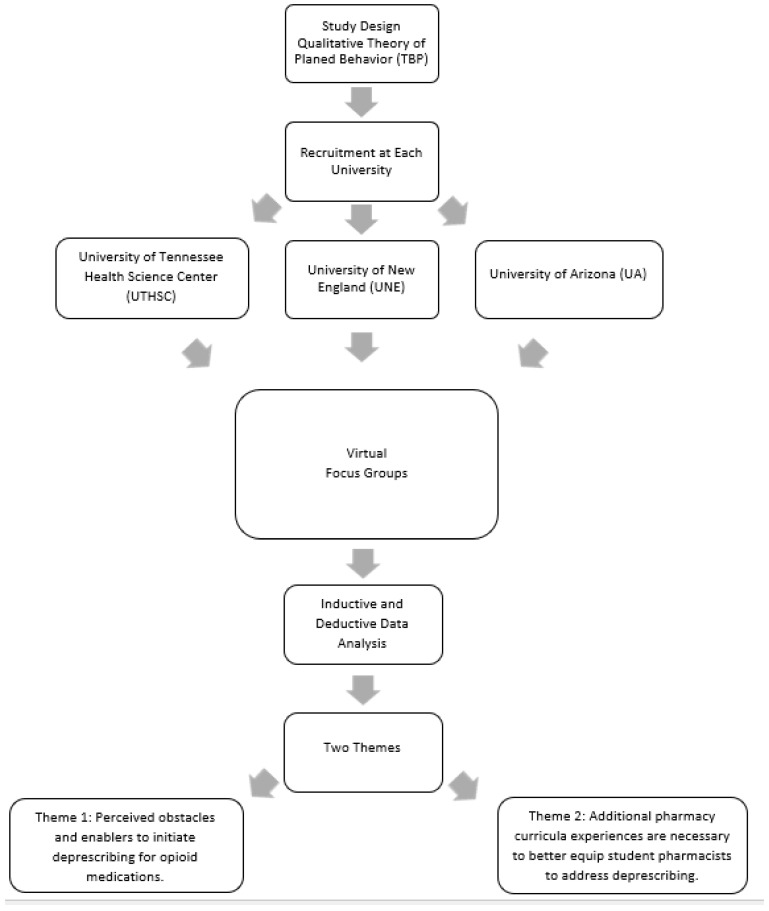
Overview of the study design, data collection, and analysis.

## Data Availability

Data is not available due to privacy and ethical restrictions.

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
