# Peer review of "Student Perspectives on the Pharmacist’s Role in Deprescribing Opioids: A Qualitative Study"

_pharmacy, 2023, doi:10.3390/pharmacy11040116_

Round 1

Reviewer 1 Report

Dear Authors,

Many thanks for writing and preparing this paper for publication.

The article good and it focusses in on a difficult scenario but a very important one and conversation we as pharmacists regularly encounter, and much of it is with the sensitivities from the patient about changing a medication they perceive to be helping hem and reducing pain.

I have a few points that I would like to comment on:

Line 85 mentions deprescribing efforts can I ask for clarity is this in relation to all medications or only opioids?

For the diagram, can I ask how many students were approached from each academic institute? and how many participated in the focus groups?  Would be good to see the uptake and involvement

Below figure 1 can the sentences starting on line 115 be moved to below the diagram to introduce the questions rather than have the diagram leading straight in to the bullet points (this could be from the typesetting)

Lines about 240 - Are there examples of OSCE looking at deprescribing other medications?

Lines about 250. Do the OSCE only concentrate of discussions between pharmacists and prescribers or are there also OCSEs practicing and role play between pharmacists and patients?

Line 264 - very good point  to increase confidence, more scenarios and increased practice.

Good identification of the 2 themes and suggestions to work on addressing these.

Many thanks for preparing and submitting the paper

Author Response

Dear Authors,

Many thanks for writing and preparing this paper for publication.

The article good and it focusses in on a difficult scenario but a very important one and conversation we as pharmacists regularly encounter, and much of it is with the sensitivities from the patient about changing a medication they perceive to be helping hem and reducing pain.

I have a few points that I would like to comment on:

Line 85 mentions deprescribing efforts can I ask for clarity is this in relation to all medications or only opioids?

Here is the paragraph: “However, researchers have found that student pharmacists lack confidence or are unprepared to engage in deprescribing efforts in practice.16 While pharmacists are well-positioned to approach opioid deprescribing with their patients, further research is needed to identify and operationalize opportunities to increase student confidence in deprescribing.”

Response: Thank you for this comment. The study cited pertained to deprescribing medications in general. Our study focuses on deprescribing opioids specifically.

For the diagram, can I ask how many students were approached from each academic institute? and how many participated in the focus groups?  Would be good to see the uptake and involvement

Response: Thank you for this valuable suggestion. The following paragraph was added to the text:” A total of 1,366 student pharmacists from three Colleges of Pharmacy were invited to participate voluntarily in the virtual FGDs (UTHSC N = 682; UNE N = 158; University of Arizona N = 526). Twenty-six student pharmacists participated in four FGDs. Of 26 participants, fourteen self-identified as male, two declined to state their gender identity, 16 self-identified as White, four as Asian, two as Latino, one as Middle Eastern, one as Pacific Islander, and one declined. Out of the 26 participants, 16 were enrolled in the fourth year, seven were enrolled in the third year, and three were enrolled in the second year.”

Below figure 1 can the sentences starting on line 115 be moved to below the diagram to introduce the questions rather than have the diagram leading straight in to the bullet points (this could be from the typesetting)

Response: Thank you for this suggestion. As we amended the text, this issue was resolved.

Lines about 240 - Are there examples of OSCE looking at deprescribing other medications?

Response: Thank you for this suggestion. We did a literature review regarding OSCEs on deprescribing; however, there are limited studies published.

Lines about 250. Do the OSCE only concentrate of discussions between pharmacists and prescribers or are there also OCSEs practicing and role play between pharmacists and patients?

Response: Thank you for this suggestion. The OSCE vary among the Colleges of Pharmacy in the US and depending on the purpose of the OSCE and what is being assessed.

Line 264 - very good point  to increase confidence, more scenarios and increased practice.

Response: Thank you for the positive feedback.

Good identification of the 2 themes and suggestions to work on addressing these.

Response: Thank you. These suggestions strengthened our manuscript.

Many thanks for preparing and submitting the paper

Reviewer 2 Report

Clear, concise, and interesting paper. The conclusion that student communication/practice skills are the barrier to engaging in deprescribing, not knowledge, is very interesting and will hopefully inform pharmacy school curriculum updates. Suitable for publication with relatively minor additions:

1. the authors provide a very sound rationale for pharmacist deprescribing of opioids. However it is less clear why and whether pharmacy students are needed to engage in deprescribing. I.e. are pharmacy students in experiential settings crucial for opioid deprescribing due to the time constraints on the licensed pharmacists/supervisors? Or is this a discussion with pharmacy students about their future practice.

2. some additional demographic/curricular details about the three pharmacy programs should be added to the methods, possibly in a table but not necessarily. Are all of these students at the same level (e.g. 3rd year)? Is the opioid and deprescribing content similar across each school curriculum? Are the experiential opportunities similar or quite different between the three schools?

Author Response

Clear, concise, and interesting paper. The conclusion that student communication/practice skills are the barrier to engaging in deprescribing, not knowledge, is very interesting and will hopefully inform pharmacy school curriculum updates. Suitable for publication with relatively minor additions:

  1. the authors provide a very sound rationale for pharmacist deprescribing of opioids. However it is less clear why and whether pharmacy students are needed to engage in deprescribing. I.e. are pharmacy students in experiential settings crucial for opioid deprescribing due to the time constraints on the licensed pharmacists/supervisors? Or is this a discussion with pharmacy students about their future practice.

Response: Thank you for this clarification.

Pharmacists play an integral role in managing a patient’s medications. Sometimes, patients may be taking more medications than they need or at too high of a dose. This is where a pharmacist recommendation as to a change in the medication regimen would be very valuable to the patient. The discussion in this manuscript focuses on preparing the student pharmacist to be able to make that recommendation, when appropriate, in their future practice. The pharmacist may initiate this change if practicing under a collaborative pharmacy practice agreement (CPPA) that authorizes them to do make the change. Not all pharmacists are practicing under a CPPA. For pharmacists who do not have a CPPA in place, then they must make the recommendation for the prescriber to decide whether to make the change.

  1. some additional demographic/curricular details about the three pharmacy programs should be added to the methods, possibly in a table but not necessarily. Are all of these students at the same level (e.g. 3rd year)? Is the opioid and deprescribing content similar across each school curriculum? Are the experiential opportunities similar or quite different between the three schools?

Response: Thank you for this clarification. We added more information in the methods about the Pharmacy curricula at the respective Colleges. We also added information about the students’ demographics.

We amended the text: The Doctor of Pharmacy (Pharm.D.) curriculum at all three Colleges (UTHSC, UNE, and the University of Arizona) is a four-year curriculum program composed of didactic classes and experiential education. Student pharmacists study pharmacotherapy, medicinal chemistry, and pharmacology of opioids throughout the Pharm.D. curriculum.

Reviewer 3 Report

This is an evaluation of pharmacy students' perspectives about opioid deprescribing. Opioid deprescribing is an important and highly relevant topic in healthcare and the pharmacist's role in opioid deprescribing can be highly prominent. In fact, the pharmacist's role in this setting probably should be more prominent than it currently is, thus the subject of this paper is important.

I do have some concerns and critiques of the paper.

Introduction:

I believe there has been over-extrapolation of general medication deprescribing literature to the context of opioid deprescribing/tapering and significant under utilization of specific opioid deprescribing literature. There has been much written about the concerns that are very specific to opioid deprescribing and these are not mentioned at all in the paper. (Consider Bohnert and Ilgen NEJM 2019;280:71-9 and Dowell et al. JAMA 2019;322:1855-6.) In addition, references are needed to support the statements in lines 69 and 70.

Methods:

I felt the methods were inadequately described. Despite the authors' citations of their previous work related to this research, I was not able to discover what I feel to be necessary information including inclusion criteria for the participant sample, sample size, and number of and distribution of representatives from each of the schools included in the study. Was saturation reached? In addition, it is not clear how the authors directed the questioning from general medication deprescribing (as is briefly described in their previously published work pertaining to the larger overarching study) to specifics about opioid deprescribing. Thus, it is not at clear whether bias was introduced and if so, how it was accounted for in the analysis. As currently written, the methods are insufficient for the study to be replicated and to determine their validity for determining and supporting the authors' conclusions.

Results:

See my above concerns about the methodology and resultant inability to confidently trust the results.

Lines 208-211 are stated like a conclusion/summary rather than a result of the research. Also, I believe it is overly-conclusive in nature based on the data presented. It's written as though it's fact, but it's not, it's based on student focus group consensus/opinion.

Author Response

This is an evaluation of pharmacy students' perspectives about opioid deprescribing. Opioid deprescribing is an important and highly relevant topic in healthcare and the pharmacist's role in opioid deprescribing can be highly prominent. In fact, the pharmacist's role in this setting probably should be more prominent than it currently is, thus the subject of this paper is important.

I do have some concerns and critiques of the paper.

Introduction:

I believe there has been over-extrapolation of general medication deprescribing literature to the context of opioid deprescribing/tapering and significant under utilization of specific opioid deprescribing literature. There has been much written about the concerns that are very specific to opioid deprescribing and these are not mentioned at all in the paper. (Consider Bohnert and Ilgen NEJM 2019;280:71-9 and Dowell et al. JAMA 2019;322:1855-6.) In addition, references are needed to support the statements in lines 69 and 70.

Response: Thank you for this suggestion. We have updated the introduction to acknowledge some of the concerns with tapering and included to 2 references suggested by the reviewer. The sentence on lines 69-70 does have a reference.

Methods:

I felt the methods were inadequately described. Despite the authors' citations of their previous work related to this research, I was not able to discover what I feel to be necessary information including inclusion criteria for the participant sample, sample size, and number of and distribution of representatives from each of the schools included in the study. Was saturation reached? In addition, it is not clear how the authors directed the questioning from general medication deprescribing (as is briefly described in their previously published work pertaining to the larger overarching study) to specifics about opioid deprescribing. Thus, it is not at clear whether bias was introduced and if so, how it was accounted for in the analysis. As currently written, the methods are insufficient for the study to be replicated and to determine their validity for determining and supporting the authors' conclusions.

Response: Thank you for this clarification. We added text in methods and results (as appropriate) to state the eligibility criteria, number of eligible students, and distribution of students from each school of pharmacy. We also added that saturation was met. We added some additional details to clarify the questioning in the focus groups.

Results:

See my above concerns about the methodology and resultant inability to confidently trust the results.

Lines 208-211 are stated like a conclusion/summary rather than a result of the research. Also, I believe it is overly-conclusive in nature based on the data presented. It's written as though it's fact, but it's not, it's based on student focus group consensus/opinion.

Response: Thank you for your suggestion. We deleted the summary sentence and transitioned into the next finding.

  1. Morse J. Strategies for sampling in: Morse j. M, editors. Qualitative nursing research: A contemporary dialogue. London: Sage Publications; 1991.
  2. Morse JM. Determining sample size. Sage Publications Sage CA: Thousand Oaks, CA; 2000.

Round 2

Reviewer 3 Report

Thank you for your response to my inquiries. The manuscript is improved and acceptable.